# Incorporation of Perillyl Alcohol into Lipid-Based Nanocarriers Enhances the Antiproliferative Activity in Malignant Glioma Cells

**DOI:** 10.3390/biomedicines11102771

**Published:** 2023-10-12

**Authors:** Tarek A. Ahmed, Alshaimaa M. Almehmady, Waleed S. Alharbi, Abdullah A. Alshehri, Fahad A. Almughem, Reem M. Altamimi, Manal A. Alshabibi, Abdelsattar M. Omar, Khalid M. El-Say

**Affiliations:** 1Department of Pharmaceutics, Faculty of Pharmacy, King Abdulaziz University, Jeddah 21589, Saudi Arabia; amnalmehmady@kau.edu.sa (A.M.A.); wsmalharbi@kau.edu.sa (W.S.A.); kelsay1@kau.edu.sa (K.M.E.-S.); 2Advanced Diagnostics and Therapeutics Institute, Health Sector, King Abdulaziz City for Science and Technology (KACST), Riyadh 11442, Saudi Arabia; abdualshehri@kacst.edu.sa (A.A.A.); falmughem@kacst.edu.sa (F.A.A.); raltamimi@kacst.edu.sa (R.M.A.); 3Healthy Aging Institute, Health Sector, King Abdulaziz City for Science and Technology (KACST), Riyadh 11442, Saudi Arabia; malshabibi@kacst.edu.sa; 4Department of Pharmaceutical Chemistry, Faculty of Pharmacy, King Abdulaziz University, Jeddah 21589, Saudi Arabia; asmansour@kau.edu.sa

**Keywords:** perillyl alcohol, lipid-based nanoformulations, docking, brain cancer, human farnesyltransferase, cell viability, industrial development

## Abstract

Perillyl alcohol (PA), a naturally existing monocyclic terpene related to limonene, is characterized by its poor aqueous solubility and very limited bioavailability. Its potential anti-cancer activity against malignant glioma has been reported. The aim was to develop PA-loaded lipid-based nanocarriers (LNCs), and to investigate their anti-cancer activity against two different brain cell lines. Non-medicated and PA-loaded LNCs were prepared and characterized. The mechanism of cytotoxic activity of PA was conducted using a molecular docking technique. The cell viabilities against A172 and ANGM-CSS cells were evaluated. The results revealed that the average particle size of the prepared LNCs ranged from 248.67 ± 12.42 to 1124.21 ± 12.77 nm, the polydispersity index was 0.418 ± 0.043–0.509 ± 0.064, while the zeta potential ranged from −36.91 ± 1.31 to −15.20 ± 0.96 mV. The molecular docking studies demonstrated that the drug had binding activity to human farnesyltransferase. Following exposure of the two glioblastoma cell lines to the PA-loaded nanoformulations, MTS assays were carried out, and the data showed a far lower half-maximal inhibitory concentration in both cell lines when compared to pure drug and non-medicated nanocarriers. These results indicate the potential in vitro antiproliferative activity of PA-loaded LNCs. Therefore, the prepared PA-loaded nanocarriers could be used to enhance drug delivery across the blood–brain barrier (BBB) in order to treat brain cancer, especially when formulated in a suitable dosage form. The size, surface charge, and lipid composition of the LNCs make them promising for drug delivery across the BBB. Detailed pharmacokinetic and pharmacodynamic assessments, including the evaluation of BBB penetration, are necessary to better understand the compound’s distribution and effects within the brain.

## 1. Introduction

Terpenes are natural organic compounds found in a wide array of plants, and their diverse chemical structures and properties make them fascinating subjects of study [1]. They consist of a number of isoperene units, a five-carbon base substance, while terpenoids are a modified type of terpene in which the compound contains oxygen [2]. Terpenes serve as the basis for the production of essential oils, flavors, and fragrances, with applications in cosmetics, perfumery, and the food industry [2]. Moreover, terpenes have shown promise in fields like medicine, where they can be harnessed for their anti-inflammatory, antimicrobial, and therapeutic properties [3]. As research in this area continues to advance, the possibilities for the practical utilization of terpene compounds from plant sources appear increasingly exciting and far-reaching. The characterization of these compounds is a multifaceted scientific endeavor, involving analytical techniques like gas chromatography–mass spectrometry (GC–MS) and nuclear magnetic resonance (NMR) spectroscopy [4]. Such methodologies enable the identification and quantification of the numerous terpenes found in different plant species, shedding light on their chemical structures and roles within ecosystems.

Perillyl alcohol (PA) is a monoterpenoid compound available in essential oils of various herbs including perilla, peppermint, sage, thyme, lemongrass, lavender, and berries [1]. PA is a small molecule (152.237 g/mole) that is slightly soluble in water (1.9 mg/mL), and has a boiling point of 119–121 °C. In terms of therapeutic activity, PA has demonstrated anti-cancer effects against lung, brain, pancreatic, mammary, and colon cancers in animal models. Various studies have investigated possible cellular targets for PA anti-proliferative activities on animal models [5]. A number of studies suggested that PA, limonene, and their metabolites interfere in the process of G protein isoprenylation, and restrict cell proliferation [6,7,8,9]. Moreover, other studies have reported that PA has an impact on the cell cycle machinery, which consequently affects the cell cycle progression of tumor cells. For example, treatment of cells in vitro leads to the downregulation of several cyclin proteins. Cyclins are the regulatory components of cyclin-dependent kinases (CDKs), the activity of which is crucial for cell cycle progression. In contrast, POH upregulates the expression of CDK-inhibitory proteins such as p15 (INK4b, CDKN2B), p21 (WAF/Cip1, CDKN1A), and p27 (Kip1, CDKN1B), all of which may suppress CDK activity [10,11,12,13]. The following cell cycle arrests restrain tumor cell progression in vitro, potentially translating into blocking of tumor growth in vivo. Furthermore, many other cellular targets of PA have been discovered, such as immediate–early genes c-Fos and c-Jun [10,14]; telomerase reverse transcriptase (hTERT) [15,16]; sodium/potassium adenosine triphosphatase (Na/K-ATPase) [17,18]; nuclear factor kappa B (NF-κB) [19,20,21]; Notch (which plays a role in tumor cell invasion and metastasis) [22]; mammalian targeting of the rapamycin complex (mTORC) [15,23]; and transforming growth factor beta (TGFβ) [10]. In addition, PA was identified as an efficient stimulant for endoplasmic reticulum (ER) stress by inducing the stimulation of ER stress markers [24]. Overall, it is not conclusive yet which mechanism has the upper hand against tumor cells, as apparently each one of these actions of PA has a role in the anti-proliferative activity. PA has been moved to clinical trials in humans after many investigations that proved its successful anti-cancer activity in pre-clinical in vitro as well as in vivo models. PA was studied in groups of patients with ovarian cancer [25], metastatic breast cancer [26], metastatic prostate cancer [27], pancreatic cancer [28], and metastatic colorectal cancer [29]. Overall, the therapeutic effect of PA was generally unimpressive. Moreover, although the GI side effects were not severe as per common clinical criteria, the chronic and unpleasant nature of these symptoms caused issues to participants to the point that some patients discontinued the therapy and stop their participation in these trials [25,27,29]. While the oral PA clinical trials have stopped and no further phase III trials have been conducted, the door is open for other potential alternative formulations types. As carriers for PA, formulations comprising a hydrogel, O/W nanoemulsion, O/W macroemulsion, and nanostructured lipid carrier were developed, and the prepared nanoformulations and the conventional hydrogel have been reported to be considered as potential vehicles in the topical delivery of PA [30]. Recently, Peczek et al. reported preparation of PA-loaded nanostructure lipid carriers to enhance drug distribution in the brain. These nanocarriers were prepared using PA as an oily phase, Gelucire^®^ 43/01 as a solid lipid, and an aqueous phase containing polysorbate 80 and soy lecithin, utilizing the hot homogenization technique [31]. Accordingly, there is a need to develop PA loaded lipid-based nanocarriers utilizing other constituents and techniques to enhance the therapeutic efficacy of PA.

In this study, we developed PA-loaded lipid-based nanocarriers utilizing a melt-emulsion and low-temperature solidification technique. The prepared nanocarriers were characterized for their size, zeta potential, and polydispersity index. Molecular docking was conducted to study the mechanism of cytotoxic activity of PA. The anti-tumor drug activity against the ANGM-CSS and A172 brain cell lines was investigated using the MTS assay. The half maximal inhibitory concentration (IC_50_) was calculated to measure the potency of PA and the prepared nanocarriers against malignant glioma.

## 2. Materials and Methods

### 2.1. Materials

Perillyl alcohol was purchased from AK Scientific (Union City, CA, USA). Tween 80 (polysorbate 80) and cholesterol were obtained from Sigma-Aldrich (St. Louis, MO, USA). Stearic acid was procured from Fischer Scientific (Loughborough, UK). Sefsol 218 was obtained from Nikko Chemicals Company, Ltd. (Tokyo, Japan). Dulbecco’s Modified Eagle Medium (DMEM), fetal bovine serum (FBS), and antibiotic solution were all obtained from Sigma-Aldrich (St. Louis, MO, USA). CellTiter 96^®^ aqueous one solution cell proliferation assay (MTS assay) was supplied by Promega (Southampton, UK). The first glioblastoma cell line used (A172) was obtained from the American Type Culture Collection under catalogue number CRL-1620^TM^ (Manassas, VA, USA). The second glioblastoma cell line (ANGM-CSS) was purchased from the European Collection of Authenticated Cell Cultures under catalogue number 08040401^TM^ (San Giovanni Rotondo, Italy).

### 2.2. Preparation of Nanocarriers

Medicated and non-medicated nanocarriers were prepared utilizing a previously mentioned melt-emulsion and low-temperature solidification technique, but with some modifications [32]. The composition of the prepared formulations is depicted in Table 1. Briefly, the calculated amounts of stearic acid and cholesterol were melted in a water bath at 80 °C. The studied oil (either PA or sefsol) was added to the melted oily mixture. An aqueous surfactant solution (1% *w*/*v*) was prepared and heated in a water bath at 80 °C. The aqueous surfactant solution was added to the oily mixture on a magnetic stirrer, at 1200 rpm, over a water bath at 80 °C. The mixture was subjected to high-speed homogenization at 23,000 rpm for 15 min. Finally, the preparation was subjected to rapid cooling by immersion into ice-cold water over a magnetic stirrer at 1200 rpm for 15 min to produce a homogenous dispersion. The prepared lipid-based nanocarriers were assigned the codes NLC1, NLC2, and NLC3.

Sefsol was incorporated into the NLC2 formulation instead of PA, in order to prepare a positive control formulation, since this oil is generally regarded as a safe category of pharmaceutical excipients [33]. A non-medicated “oil-free” formulation, namely NLC3, was also prepared.

### 2.3. Characterization of the Prepared Nanocarriers

The obtained formulations were characterized for particle size, polydispersity index (PDI), and surface charges (zeta potential) using a Malvern Zetasizer Nano ZSP, Malvern Panalytical Ltd. (Malvern, UK). Each formulation was diluted with distilled water in a ratio of 1:5 (*v*/*v*), to ensure low density of the nanocarriers in the sample compartment, which limits interaction between the particles [34]. The Malvern Zetasizer Nano ZSP utilizes the dynamic light scattering (DLS) technique to determine the particle sizes of nanoformulations. This technique involves measuring the fluctuations in scattered light caused by the Brownian motion of particles suspended in a liquid medium. Additionally, the instrument calculates the PDI from the data obtained during the particle size measurement. The zeta potential is a measure of the electrostatic charge at the surface of nanocarriers. It provides insights into the stability and colloidal behavior of nanoformulations. The electrophoretic mobility of particles in an electric field is used to calculate the zeta potential. The zeta potential of the particles was measured at a scattering angle of 13°, an equilibration time of 300 s, and a temperature of 25 °C. The instrument applies an electric field to the sample and measures the velocity of particles as they move through the field. For each sample, the number of runs, scans, voltage, and attenuation settings were automatically determined. The average value of three readings was taken.

The morphological characterization of the PA-loaded lipid based nanocarriers formulation was conducted using transmission electron microscopy (TEM). A JEOL JEM-2100 TEM from Tokyo, Japan, was used. A small volume of the formulation was dispersed onto a carbon support film TEM grid. Then, the sample was air-dried, loaded in the sample holder, and images were captured.

### 2.4. Molecular Docking Study

The mechanism of the antitumor activity of perillyl alcohol, and to a lesser extent its precursor limonene, involves inhibiting the process of isoprenylation of small G proteins, including Ras oncoprotein activity through inhibition of the farnesylation of p21-Ras [35,36]. Ras genes are the most frequently mutated oncogenes in human cancer that are required for normal activity. To further explain the mechanism of PA’s cytotoxic activity, we conducted a molecular docking experiment using the Schrodinger program (Schrödinger Release 2022-3: Schrödinger, LLC, New York, NY, USA, 2021).

#### 2.4.1. Preparation of Protein

The crystal structure of human farnesyltransferase with farnesyl diphosphate (PDB ID: 1LDZ) [37] from the protein databank was downloaded, and then processed with the Protein Preparation Wizard in Maestro Schrödinger [38]. The alterations made involved adding the missing hydrogens to the residues, correcting the metal ionization states, remoting water molecules that were further than 5 Å, assigning proper charges, and allowing the protein to go through restrained minimization with OPLS4 as the force field.

#### 2.4.2. Ligand Preparation

The two compounds, perillyl diphosphate and farnesyl diphosphate, were made ready for docking by converting their two-dimensional structures to three-dimensional ones, and minimizing the energy of each employing the OPLS3 force field. Then, hydrogen atoms were added, and all of the conceivable protonation states and tautomeric forms for each molecule were generated with a pH of 7.0 ± 0.2 using the Epik tool, with the desalt function selected. In order to optimize any existing hydrogen bonds, the pKa of any ionizable groups was forecasted using the PROPKA tool [39].

#### 2.4.3. Grid Generation and Molecular Docking

A grid box was generated in the vicinity of the human farnesyltransferase active site (1LDZ) co-crystallized with farnesyl diphosphate, implemented with the “Receptor-Grid-Generation” feature of the Schrödinger suite [40]. The box structure was based off the centroid of the workspace ligand, with an automatic X, Y, Z dimension length of 10 Å. Then, perillyl diphosphate was docked inside the box, conducted once with standard precision (SP) and thrice with extra precision (XP) settings. All of the default parameters were implemented [41], apart from the VdW radii scaling factor and partial charge cutoff being set to 1.0 and 0.25, respectively. Then, the “Ligand Docking” tool was employed for docking [40]. To validate the docking method, farnesyl diphosphate was re-docked inside the grid box and evaluated. The evaluation comprised three application-specific scores, the gscore (for ranking the different compounds), the emodel score (for ranking the different conformers), and the XP gscore. Glide utilizes emodel scoring to select the optimal poses of the docked compounds and ranks them via the gscore. The XP gscore, on the other hand, ranks the poses generated by XP Glide mode, which takes into account major binding forces and structural motifs influencing the binding affinity by the use of equations [41].
XP Glide Score = E*_coul_* + E*_VdW_* + E*_bind_* + E*_penalty_*
E_bind_ = E_hyd_enclosure_ + E_hb_nn_motif_ + E_hb_cc_motif_ + E_PI_ + E_hb_pair_ + E_phobic_pair_
E_penalty_ = E_desolv_ + E_ligand_strain_

For each of the following descriptors, the energy (E) was calculated: Coulomb energy (E_coul_), van der Waals energy (E_VdW_), energy that promotes binding (E_bind_), energy that penalizes binding (E_penalty_), hydrophobic enclosure (E_hyd_enclosure_), special neutral-neutral hydrogen-bond motifs (E_hb_nn_motif_), special charged-charged hydrogen-bond motifs (E_hb_cc_motif_), pi–cation interactions (E_PI_), hydrogen bond pairs (E_hb_pair_), lipophilic pairs (E_phobic_pair_), and desolvation energy (E_desolv_) [42].

### 2.5. Determination of the Half Maximal Inhibitory Concentration “IC_50_”

The in vitro cell viability of the free drug and the prepared nanocarriers was evaluated using the colorimetric assay, MTS, following a 24-hour cell exposure of formulations to the A172 and ANGM-CSS cells. The in vitro metabolic activity assessments of the free drug and NLC formulations were performed using the MTS assay, according to the modified method published by Alzahrani et al. [43]. The culturing of the utilized human glioblastoma cells was routinely maintained in DMEM, supplemented with streptomycin 100 μg/mL, penicillin 100 U/mL, and 10% (*v*/*v*) fetal bovine serum (FBS). The cells were harvested using trypsin and counted with the trypan blue exclusion test, followed by seeding 1.5 × 10^4^ cells per well into 96-well plates. The cells were incubated overnight in a cell culture incubator at 37 °C and 5% CO_2_. A volume of 100 μL of increasing concentration of the free drug, from 7.81 to 32,000 µg/mL, and the tested nanocarriers, from 7.81 to 500 µg/mL, was then exposed to the human cells for 24 h. Cells incubated with 0.2% Triton X-100 were used as a positive control to disrupt the cell membrane through solubilization of membrane lipids and proteins, leading to cell lysis [44]; meanwhile, cells incubated with DMEM only were used as a negative control. The consumed medium was aspirated from each well, then 100 μL of fresh DMEM was added, followed by 20 μL of the MTS reagent. The cells were incubated for a further 3 h at 37 °C and 5% CO_2_. A Cytation 3 absorbance microplate reader (BIOTEK Instruments Inc., Winooski, VT, USA) was used at 492 nm to measure the ability of the cells to produce formazan color upon the living, and the cellular viability (%) was calculated with the following equation:Cellular Viability (%) = (S − T)/(H − T) × 100
where S is the absorbance of the cells treated with the applied formulations, T is the absorbance of the cells treated with Triton X-100, and H is the absorbance of the cells treated with DMEM. The results are presented as the mean ± SD of at least three independent measurements. Then, the half-maximal cell growth inhibitory concentration (IC_50_) was calculated for all of the applied samples.

### 2.6. Statistical Analysis

The IC_50_ values were calculated using an online tool: AAT Bioquest, Inc., (Pleasanton, CA, USA); Available online: https://www.aatbio.com/tools/ic50-calculator (accessed on 8 January 2023).

## 3. Results and Discussion

### 3.1. Characterization of the Prepared Nanocarriers

Nanostructure lipid carriers are part of a novel drug delivery system that has potential formulations in the cosmetics and pharmaceutical markets due to their biocompatibility, non-toxicity, high drug loading, stability, and improving bioavailability [45]. In this study, PA-loaded nanostructure lipid carriers were developed using the melt-emulsion and low-temperature solidification technique.

Stearic acid and cholesterol are commonly chosen components in formulating lipid-based nanocarriers due to their unique properties that are essential for the success of these drug delivery systems. Stearic acid, a hydrophobic molecule, is employed to create the core of the lipid-based nanocarrier formulation. Stearic acid is typically a solid at room temperature, contributing to the structural integrity of LNPs [46]. Cholesterol is incorporated to decrease the drug leakage from the prepared nanocarriers, as it increases the membrane rigidity [47]. Cholesterol also decreases the permeability of the lipid bilayer and enhances the biocompatibility of LNPs, which makes them more suitable for use in biological systems. It reduces the chances of immune system recognition and clearance, thus increasing the circulation time of LNPs in the bloodstream. The concentrations of stearic acid, cholesterol, and Tween 80 were selected based on the optimization of these component levels in our previously published research during the development of solid lipid nanocarriers [32,48]. A graphical representation of the prepared drug-loaded nanostructure lipid carrier is illustrated in Figure 1. The prepared medicated and non-medicated nanocarriers were characterized for their size, PDI, and zeta potential. The average particle size ranged from 248.67 ± 12.42 to 1124.21 ± 12.77 nm; the obtained values for the PDI were 0.418 ± 0.043–0.509 ± 0.064, while the zeta potential was between −36.91 ± 1.31 and −15.20 ± 0.96 mV. Incorporation of the oil (either PA or sefsol) resulted in a marked decrease in the particle size, from 1124.21 ± 12.77 nm in NLC3 to 330.47 ± 22.11nm in NLC1 and 248.67 ± 12.42 nm in NLC2. The effect may be attributed to the crystalline solid lipid core of the NLC3 formulation when compared to the amorphous lipid core of the NLC1 and NLC2 formulations. This explanation was previously mentioned by Aditya et al. during an investigation into the effect of the physical state and composition of the incorporated lipid substances on the formation and characterization of lipid nanocarriers [49]. All of the prepared nanocarriers showed negatively charged particles, an effect that may be attributed to the presence of stearic acid and cholesterol in all of the prepared nanocarriers. It was previously mentioned that the preferred zeta potential value to obtain stable particles should be >±30 [50]. Although the prepared nanocarriers have zeta potential values that are lower than the recommended value, these values are sufficient to stabilize the prepared nanocarriers by both electrostatic and steric stabilization [51]. Thermal and long-term storage stability testing are recommended to assess the behavior of these nanocarriers under different conditions. Finally, the prepared nanocarriers demonstrated good homogeneity, since they showed a PDI value lower than 0.7. A PDI value greater than 0.7 is an indication of a broad particle size distribution, as reported by Danaei et al. [52]. The PDI is very important in certain applications, such as cellular uptake [52], targeting specific receptors [53], or crossing biological barriers [54]. A narrow size distribution may be critical to ensure consistent and efficient interactions with biological systems, and hence a positive impact on the clinical effectiveness of the active pharmaceutical ingredient [55].

The size of lipid-based nanocarriers that have the ability to cross the blood–brain barrier (BBB) can vary depending on the specific formulation and design of the nanocarrier. However, in general, nanocarriers for drug delivery to the brain are typically designed to be within the nanometer range. This is because the BBB is highly selective and restricts the passage of larger molecules and particles [56]. Smaller molecules, such as nanocarriers with diameters less than 100 nanometers, cross the BBB through a concentration gradient with the assistance of passable transporters, while larger molecules (such as peptides and proteins) may rely on an endocytic mechanism or other active transport mechanisms to facilitate their entry into the brain [57]. Accordingly, the three prepared formulations, NLC1, NLC2, and NLC3, demonstrated sizes of 330.47 ± 22.11, 248.67 ± 12.42, and 1124.21 ± 12.77 nm, respectively, and were expected to cross the BBB through an endocytic mechanism. Additionally, the passage of the PA-loaded lipid-based nanocarrier formulation via a concentration gradient is also possible, since this formulation illustrated a size of 100–160 nm, as shown in the TEM image of Figure 2. Optimization of the formulation and processing parameters and surface modification of the nanocarrier surface will be conducted in our future research. Administration of this nanocarrier via a nasal gel could be another way for brain delivery, as previously described in our research [58].

A TEM image for our drug-loaded lipid-based nanocarrier formulation is illustrated in Figure 2. The results demonstrate that the studied lipid-based nanocarrier formulation is spherical in shape. The formulation exhibits a size that is lower than that obtained using DLS, an effect that may be attributed to sample dilution and proper dispersion prior to the sample being mounted on the grid, which prevent particle agglomeration [59].

### 3.2. Molecular Docking Studies Analysis

Farnesyl alcohol is a type of organic compound known as an alcohol, and is linked with the therapeutic potential of human farnesyltransferase (FTase). FTase is an enzyme that is found in all eukaryotic cells, and its primary function is to catalyze the transfer of farnesyl moieties onto proteins, thus playing an essential role in the post-translational modification of proteins required for their function [60]. FTase inhibitors and activators can be used to modulate the activity of this enzyme. Farnesyl alcohol can be used to activate FTase, and it has been used to successfully regulate different physiological processes and pathways, with evidence showing that it can be used for treating various cancer types, immune and metabolic disorders, and pathogenic infections [61].

Perillyl alcohol and farnesyl alcohol are both monoterpenoids, and are related compounds that exhibit a molecular bridge between cyclopentane and cyclohexane. The major difference between the two compounds is that perillyl alcohol contains a cyclopentyl group, while farnesyl alcohol contains a cyclohexyl group. After preparing FTase protein and ligands, the 3D structures were loaded into the designated grid box in the prepared protein (PDB-ID: 1LD7). FTase natively binds to farnesyl diphosphate, as well as to perillyl diphosphate, which were both placed into the co-crystallized active site of the farnesyltransferase. Table 2 lists the docking scores determined from the minimum XP scores. Calculations of several docking scores were carried out, including the Glide e-model, the Glide gscore, and the XP gscore. The XP and Glide docking scores demonstrated that perillyl diphosphate had binding activity to FTase, with a score of −9.628 kcal/mol, in comparison to farnesyl diphosphate’s score of −11.963 kcal/mol (Figure 3).

The binding affinity of perillyl diphosphate with the protein was scrutinized. To corroborate these results, the natural ligand farnesyl diphosphate was redocked next to the perillyl diphosphate, and the positions of the docked molecules were studied. Figure 4 shows the bonding properties of the native ligand and FTase from 3D and 2D perspectives. Both perillyl and farnesyl diphosphate have the same binding pattern with FTase, as they bind with Arg 791 and Lys 794 via ionic interactions, and through hydrogen bonds with His 748, Arg 791, Lys 794, and Tyr 800; meanwhile, the hydrophobic parts of both molecules bind with Tyr 751, Cys 754, and Trp 803.

Redocking studies were carried out by docking the co-crystallized ligand (farnesyl diphosphate) back into the protein, in order to verify the accuracy of the docking process. The primary goal was to evaluate the predicted binding posture’s accuracy by comparing it with the farnesyl diphosphate crystallographic position. The anticipated binding pose has a remarkable low root mean square deviation value of 0.9941, and closely approximates the crystallographic pose. This observation firmly establishes a high degree of agreement between farnesyl diphosphate’s expected and actual binding interactions (Figure 5).

### 3.3. Antiproliferative Activity against ANGM-CSS and A172 Cells

The assessment of drug formulation efficacy on living cancerous cells is an important step to prove its suitability for therapeutic applications. Different doses of PA and NLC formulations exposed to A172 and ANGM-CSS cells (Figure 6A,B) were evaluated over a 24-hour period, and the results are summarized in Table 3.

Increasing the concentration of PA essential oil exhibited high antiproliferative activity for both of the glioblastoma cell lines at only the highest dose applied (32,000 µg/mL). This is inconsistent with a previous study on brain cancer cells that reported the application of PA inhibits the viability of glioblastomas in a dose-dependent manner [62]. The IC_50_ values for the PA drug were calculated to be 21,449.449 µg/mL and 22,036.925 µg/mL for the A172 and ANGM-CSS cells, respectively. The loading of PA into the nanocarrier delivery system (NLC1 formulation) showed a far lower IC_50_ value of 109.717 µg/mL in the A172 cells, and 110.576 µg/mL in the ANGM-CSS cells when compared to the sefsol-based formulation (NLC2) and the positive control non-medicated nanocarriers (NLC3). The NLC2 formulation revealed a higher IC_50_ of 274.671 µg/mL on the A172 cells, and an IC_50_ of 287.126 µg/mL on the ANGM-CSS cells following the incubation of the formulations for 24 h. The LNC3 formulation demonstrated IC_50_ values of 353.24 and 335.58 µg/mL, respectively, on the same cells as depicted in Table 3. These results suggest that PA-loaded nanocarriers can be considered as potential anti-cancer therapy for glioblastoma at the in vitro level, and that further in vitro and in vivo studies are important prior to claiming the suitability of this delivery system for brain therapy.

It must be mentioned that while non-medicated nanocarriers do not contain therapeutic agents, they can still exhibit anti-cancer activity due to their accumulation within the cancer cells, owing to the permeation and retention effect [63]; this effect allows nanocarriers to stay in the tumor tissue longer, increasing their chances of interacting with cancer cells.

Therefore, the overall results indicate the role of lipid-based nanocarriers toward improving the solubility of PA through encapsulation of the drug in the nanocarriers’ core, which minimizes the exposure of PA to water, preventing its aggregation or precipitation. By encapsulating PA within these nanocarriers, it becomes possible to enhance its solubility and overall therapeutic potential.

## 4. Conclusions

In this research, three lipid-based nanocarriers containing stearic acid and cholesterol as the main components were prepared utilizing a melt-emulsion and low-temperature solidification technique. It was observed that including essential oils, either PA or sefsol, reduces the size and the intensity of the negative charge, as observed in the NLC1 and NLC2 formulations, in comparison to NLC3. In terms of the docking site, PA showed a binding activity to human farnesyltransferase. Thereafter, the antiproliferative activity of the prepared formulations was evaluated in two cancerous glioblastoma cell lines, which are the A172 and ANGM-CSS cells. This was performed by measuring the IC_50_ values after MTS assay evaluation. It was clearly observed that both NLC1 and NLC2 have a lower IC_50_ in both cell lines compared to NLC3, which does not contain any essential oil. Interestingly, the IC_50_ of NLC1, which contains PA, was lowered to more than half the IC_50_ of NLC2, which contains sefsol as a positive control in both cell lines. Together, this research provides a novel delivery platform of PA-loaded nanocarriers with the potential to be administered for the treatment of brain cancer after loading into a suitable dosage form such as intranasal gels, solutions, or suspensions. This platform has the advantage of overcoming existing challenges of free drugs, including crossing the blood–brain barrier. Moreover, these nanocarriers can be incorporated into other dosage forms, such as creams, gels, suspensions, or oral capsules, making them adaptable to different routes of administration and patient preferences. This adaptability not only improves patient compliance, but also opens up opportunities for targeted and localized drug delivery, reducing side effects, and enhancing overall treatment outcomes. Future in vivo studies are warranted to demonstrate the efficacy of the formulated nanocarriers, and to investigate the pharmacokinetics after loading of the prepared drug nanocarrier in a suitable dosage form.

## Figures and Tables

**Figure 1 biomedicines-11-02771-f001:**
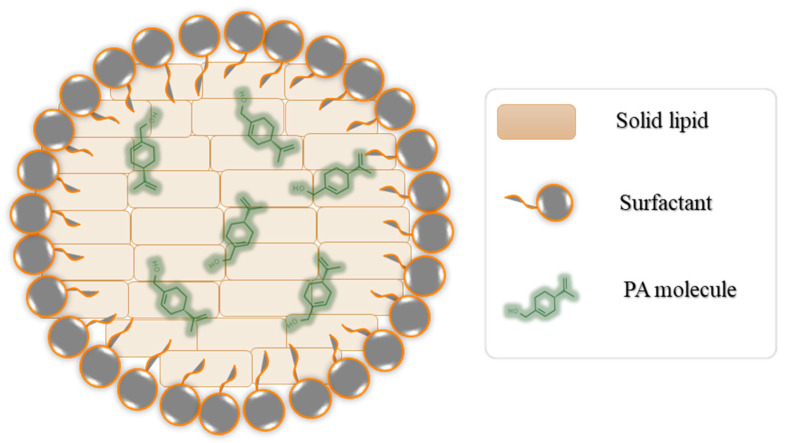
Graphical representation of PA-loaded lipid-based nanocarrier formulation.

**Figure 2 biomedicines-11-02771-f002:**
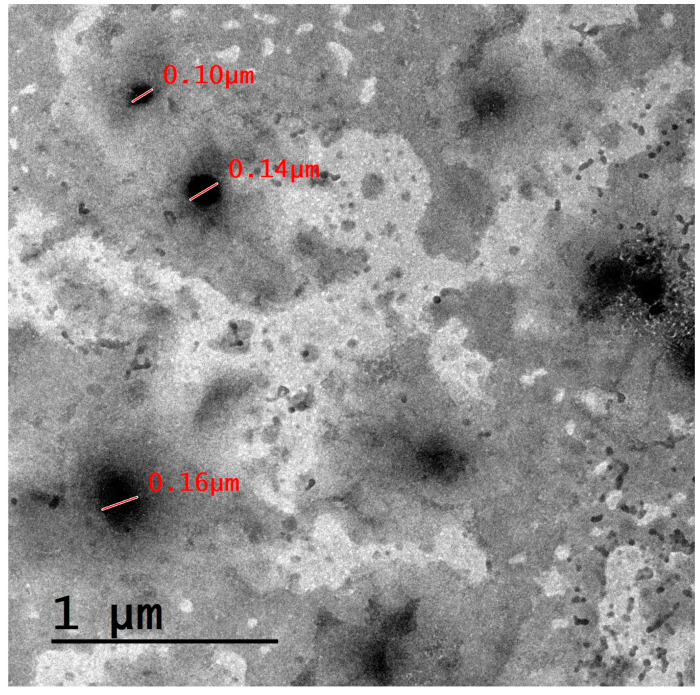
TEM micrograph for the PA-loaded lipid-based nanocarrier formulation.

**Figure 3 biomedicines-11-02771-f003:**
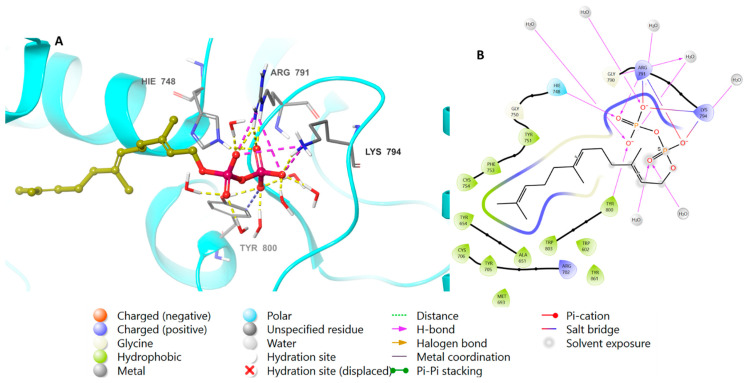
Molecular docking of farnesyl diphosphate in FTase (PDB: 1LD7). (**A**), 3D representation of farnesyl diphosphate in golden color within the active site, showing the H-bond (dashed yellow) and ionic interaction (dashed pink). (**B**), 2D representation of binding interactions of farnesyl diphosphate with amino acid residues in the active site within a 4 Å distance.

**Figure 4 biomedicines-11-02771-f004:**
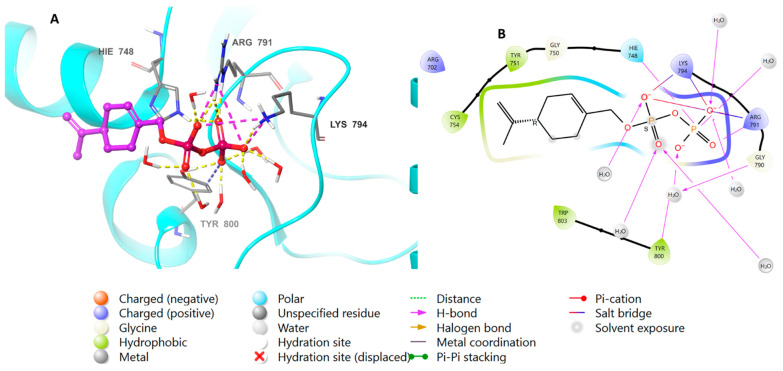
Molecular docking of perillyl diphosphate in FTase (PDB: 1LD7). (**A**), 3D representation of perillyl diphosphate in pink color within the active site, showing the H-bond (dashed yellow) and ionic interaction (dashed pink). (**B**), 2D representation of binding interactions of perillyl diphosphate with amino acid residues in the active site within a 4 Å distance.

**Figure 5 biomedicines-11-02771-f005:**
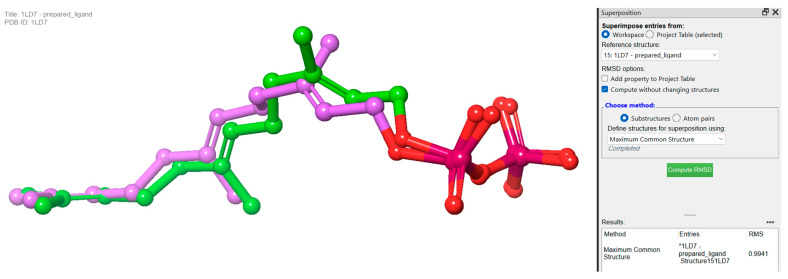
Redocking experiments to validate docking accuracy. Co-crystallized ligand (farnesyl diphosphate) was redocked into the prepared protein, comparing predicted binding pose (pink) with crystallographic pose (green) of farnesyl diphosphate.

**Figure 6 biomedicines-11-02771-f006:**
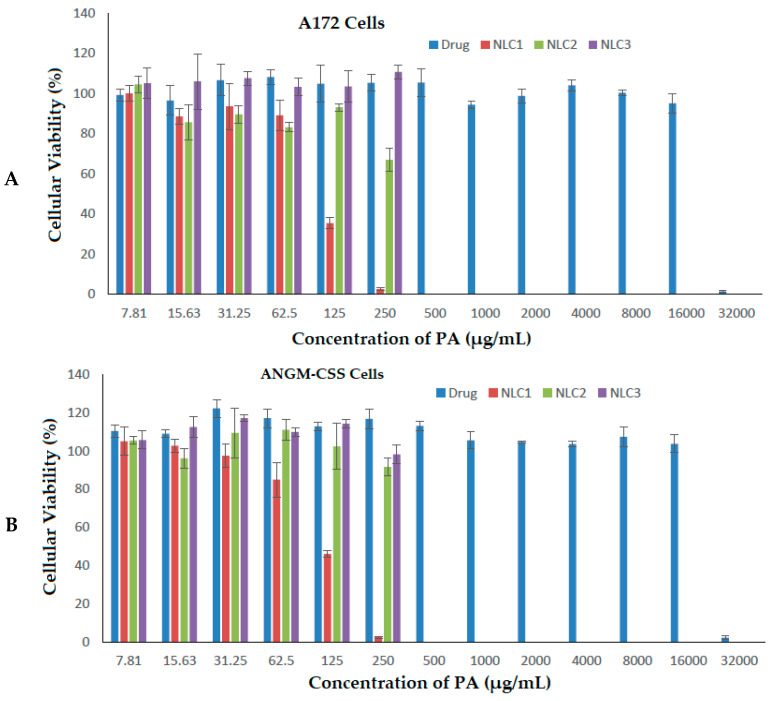
Cell viabilities of A172 cells (**A**) and ANGM-CSS cells (**B**) after treatment with free drug and the prepared nanoformulations.

**Table 1 biomedicines-11-02771-t001:** Composition and characterization of the prepared nanocarriers.

Formulation	NLC1	NLC2	NLC3
Composition			
% of Stearic acid (*w*/*v*)	1.272	1.272	1.272
% of Cholesterol (*w*/*v*)	0.16	0.16	0.16
Sefsol (mg/mL)	-	5	-
PA (mg/mL)	5	-	-
% of Tween 80 in aqueous phase (*w*/*v*)	1	1	1
Characterization			
Size (nm)	330.47 ± 22.11	248.67 ± 12.42	1124.21 ± 12.77
PDI	0.509 ± 0.064	0.504 ± 0.065	0.418 ± 0.043
Zeta potential (mV)	−15.20 ± 0.96	−17.76 ± 1.12	−36.91 ± 1.31

Abbreviations: NLC, nanostructure lipid carrier; PA, perillyl alcohol; PDI, polydispersity index.

**Table 2 biomedicines-11-02771-t002:** Docking scores determined from the minimum XP scores.

Compounds	Docking Score	XP Gscore	Glide Gscore	Glide Emodel
Farnesyl diphosphate	−11.963	−11.963	−11.963	−142.926
Perillyl diphosphate	−9.54	−9.628	−9.628	−124.045

**Table 3 biomedicines-11-02771-t003:** IC_50_ values for free drug and NLC formulations obtained upon 24-hour exposure of the glioblastoma cell lines A172 and ANGM-CSS.

Formulations	IC_50_ (µg/mL)	
	A172 Cells	ANGM-CSS Cells
Free drug	21,449.449	22,036.925
NLC1	109.717	110.576
NLC2	274.671	287.126
NLC3	353.244	335.586

## Data Availability

Not applicable.

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
