# Peer review of "Incorporation of Perillyl Alcohol into Lipid-Based Nanocarriers Enhances the Antiproliferative Activity in Malignant Glioma Cells"

_biomedicines, 2023, doi:10.3390/biomedicines11102771_

Round 1
Reviewer 1 Report
Here Ahmed et al described the incorporation of Perillyl alcohol (PA) into solid lipid nanoparticles (SLNP) for eventual therapy for Glioblastoma (GBM). Besides conducting physical characterization of nano formulation, the authors conducted molecular docking study to target Farnesyltransferase and in vitro efficacy study in two GBM cell lines.
While the idea to package PA into nanoformulation is promising, there are glaring issues in the current article. I would not suggest the acceptance of this article due to the issues below:
- The title of the article is misleading. The molecular docking study has minimal relation to the proposed idea and study design. It is incoherent with the rest of the article. It would be more acceptable if the authors conducted simulation to evaluate PA incorporation & release from SLNP.
- Considering the size of SLNP, it’s very unlikely to cross BBB and reach target GBM cells.
- IC50 of unloaded SLNP is somewhat comparable to proposed SLNP1, especially as compared to non-formulated PA. The level of nonspecific toxicity is not acceptable.
- Related to the two points above, I would recommend the authors to try other SLNP instead or at least show clearer window to prove the proposed idea
Author Response
Reviewer 1
Here Ahmed et al described the incorporation of Perillyl alcohol (PA) into solid lipid nanoparticles (SLNP) for eventual therapy for Glioblastoma (GBM). Besides conducting physical characterization of nano formulation, the authors conducted molecular docking study to target Farnesyltransferase and in vitro efficacy study in two GBM cell lines.
While the idea to package PA into nanoformulation is promising, there are glaring issues in the current article. I would not suggest the acceptance of this article due to the issues below:
- The title of the article is misleading. The molecular docking study has minimal relation to the proposed idea and study design. It is incoherent with the rest of the article. It would be more acceptable if the authors conducted simulation to evaluate PA incorporation & release from SLNP.
Reply
Upon your request, the title of the article has been changed into “Incorporation of Perillyl Alcohol into Lipid-based Nanocarriers Enhances the Antiproliferative Activity in Malignant Glioma Cells”.
- Considering the size of SLNP, it’s very unlikely to cross BBB and reach target GBM cells.
Reply
The size of lipid-based nanocarriers that have the ability to cross the blood-brain barrier (BBB) can vary depending on the specific formulation and design of the nanocarrier. However, in general, nanocarriers for drug delivery to the brain are typically designed to be within the nanometer range. This is because the BBB is highly selective and restricts the passage of larger molecules and particles [Reference # 56]. Smaller molecules, such as nanoparticles with diameters less than 100 nanometers, cross the BBB through a concentration gradient with the assistance of passable transporters, while larger molecules (such as peptides and proteins) may rely on endocytic mechanism or other active transport mechanisms to facilitate their entry into the brain [Reference # 57]. Accordingly, the prepared three formulations; NLC1, NLC2, and NLC3 that demonstrated a size of 330.47±22.11, 248.67±12.42, and 1124.21±12.77 nm, respectively expected to cross the BBB through endocytic mechanism. Additionally, passage of the PA-loaded lipid-based nanocarriers’ formulation via concentration gradient is also possible since this formulation illustrated a size 100-160 nm as shown in the TEM image of Figure 2. Optimization of the formulation and processing parameters and surface modification of the nanocarrier surface will be conducted in our future work. Administration of this nanocarrier in a nasal gel could be another way for brain delivery as previously described in our work [Reference # 58].
This explanation has been added to the modified manuscript.
- IC50 of unloaded SLNP is somewhat comparable to proposed SLNP1, especially as compared to non-formulated PA. The level of nonspecific toxicity is not acceptable.
Reply
We apologize for the confusion in the text of the last version of the manuscript. We have corrected the text to make it more clear for the reader.
Moreover, while non-medicated nanoparticles do not contain traditional drugs or therapeutic agents, they can still exhibit anticancer activity due to their accumulation within the cancer cells owing to the permeation and retention effect (Reference # 63) that allows nanoparticles to stay in the tumor tissue longer, increasing their chances of interacting with cancer cells.
This explanation has been added to the revised manuscript.
- Related to the two points above, I would recommend the authors to try other SLNP instead or at least show clearer window to prove the proposed idea
Reply
We have modified the manuscript title, included more details supported with references, and modified the name of the prepared nanocarrier to address the reviewer comments.
Reviewer 2 Report
The manuscript by Tarek Ahmad and colleagues presents a comprehensive study on the development of PA-loaded solid lipid nanoparticles (SLNPs) and their anti-cancer activity against two different brain cell lines. The research delves into the potential anticancer activity of Perillyl alcohol (PA) against malignant glioma and its challenges due to poor solubility and limited bioavailability. The study showcases the preparation and characterization of SLNPs, their molecular docking technique, and their cytotoxic activity. The results indicate that PA-loaded SLNPs have potential in vitro antiproliferative activity, suggesting their potential for drug delivery across the blood-brain barrier (BBB) to treat brain cancer.
Major Comments:
1. While the in vitro results are promising, it is crucial to demonstrate the efficacy of the PA-loaded SLNPs in vivo. This would provide a more comprehensive understanding of the potential therapeutic benefits and safety profile of the Nano formulation.
Minor Comments:
1. It would be beneficial to provide more detailed information on the characterization techniques used for the SLNPs. This would give readers a clearer understanding of the methods employed.
2. Given that one of the challenges with PA is its poor solubility, a more in-depth discussion on how the SLNPs improve the solubility of PA would be valuable.
3. The manuscript mentions the use of stearic acid and cholesterol as main components in the lipid-based nano-formulations. A clearer explanation of the choice of these lipids and their role in the formulation would enhance the manuscript.
4. The manuscript mentions the potential of the SLNPs when formulated in a suitable dosage form. It would be beneficial to discuss potential dosage forms and their implications for drug delivery and patient compliance.
Author Response
Reviewer 2
The manuscript by Tarek Ahmad and colleagues presents a comprehensive study on the development of PA-loaded solid lipid nanoparticles (SLNPs) and their anti-cancer activity against two different brain cell lines. The research delves into the potential anticancer activity of Perillyl alcohol (PA) against malignant glioma and its challenges due to poor solubility and limited bioavailability. The study showcases the preparation and characterization of SLNPs, their molecular docking technique, and their cytotoxic activity. The results indicate that PA-loaded SLNPs have potential in vitro antiproliferative activity, suggesting their potential for drug delivery across the blood-brain barrier (BBB) to treat brain cancer.
Major Comments:
- While the in vitro results are promising, it is crucial to demonstrate the efficacy of the PA-loaded SLNPs in vivo. This would provide a more comprehensive understanding of the potential therapeutic benefits and safety profile of the Nano formulation.
Reply
We completely agree with the reviewer points of view on conducting in vivo experiment which will be our focus in the upcoming work. We have mentioned in the modified manuscript that in vivo studies are warranted to demonstrate the efficacy of the formulated nanoparticles and to investigate the pharmacokinetics after loading of the prepared drug nanocarrier in a suitable dosage form.
Minor Comments:
- It would be beneficial to provide more detailed information on the characterization techniques used for the SLNPs. This would give readers a clearer understanding of the methods employed.
Reply
More detailed information have been added to the manuscript.
- Given that one of the challenges with PA is its poor solubility, a more in-depth discussion on how the SLNPs improve the solubility of PA would be valuable.
Reply
More discussion about the role of lipid-based nanoparticles on impriving the drug solubility has been added to the revised manuscript.
- The manuscript mentions the use of stearic acid and cholesterol as main components in the lipid-based nano-formulations. A clearer explanation of the choice of these lipids and their role in the formulation would enhance the manuscript.
Reply
An explanation for the role of stearic acid and cholesterol as main components in the formulation has been added.
- The manuscript mentions the potential of the SLNPs when formulated in a suitable dosage form. It would be beneficial to discuss potential dosage forms and their implications for drug delivery and patient compliance.
Reply
Potential dosage forms for lipid-based nanoparticles and their implications have been added to the revised manuscript.
Reviewer 3 Report
In this paper, the authors present the results of preparation of solid lipid nanoparticles loaded with PA and subsequent study of their anticancer activity against two different brain cell lines. The article is rather brief, the introduction does not sufficiently disclose the current state of research on this topic. In addition, the article has a number of remarks that require correction:
1) The authors begin the introduction by characterizing and classifying essential oils, substances contained in them, as well as the classification of phytochemical compounds. However, taking into account the large amount of known information about them, it would be more logical to begin the article immediately with the characterization of terpene compounds of plant origin and the possibilities of their practical use.
2) In the article the authors actively use the term nanoparticles, although taking into account the characteristics (composition, size) of the materials obtained by them it would be more correct to call them micelles or nanoemulsions. In addition, it is extremely desirable for proof of nanoscale materials to cite TEM data, in this case probably in its cryo-variant. Please take this remark into account in your further work with the manuscript.
3) There is a rather large number of works devoted to the synthesis of nanostructured lipid carriers (NLCs) containing perillyl alcohol, but the authors do not raise this when discussing the current state of the problem. The authors should revise the introduction and present data on the synthesis of perillyl alcohol-based micelles and testing of the resulting materials as potential antitumor agents.
4) The authors should unify terminology, in particular the spelling of their research objects (nanoparticles, nanodelivery agents, nanoemulsions, etc.).
5) All abbreviations (in particular, NLC) should be decoded at their first use in the text (with a dash, please).
6) The size of the order of 270-1120 nm does not allow to attribute the materials obtained by the authors to nanostructured materials.
7) Micelle size was measured by DLS method at 25 C°. Whereas the cytotoxicity was measured at 37 °C. Taking into account the expressed potential of application of the obtained materials in biomedical fields it is necessary to be oriented on conditions of testing of their characteristics similar to biomimetic ones, in particular it is necessary to supplement the data on particle size, polydispersity index and zeta potential with measurements obtained at 37 °C.
8) The authors should provide an explanation of the choice of Triton X as a positive control for the determination of the half maximal inhibitory concentration "IC50"
9) The authors should provide data on the stability of the resulting emulsions.
A number of terms require checking of spelling.
Author Response
Reviewer 3
In this paper, the authors present the results of preparation of solid lipid nanoparticles loaded with PA and subsequent study of their anticancer activity against two different brain cell lines. The article is rather brief, the introduction does not sufficiently disclose the current state of research on this topic. In addition, the article has a number of remarks that require correction:
- The authors begin the introduction by characterizing and classifying essential oils, substances contained in them, as well as the classification of phytochemical compounds. However, taking into account the large amount of known information about them, it would be more logical to begin the article immediately with the characterization of terpene compounds of plant origin and the possibilities of their practical use.
Reply
We appreciate the reviewer comment. We have included the requested details in the revised manuscript.
- In the article the authors actively use the term nanoparticles, although taking into account the characteristics (composition, size) of the materials obtained by them it would be more correct to call them micelles or nanoemulsions. In addition, it is extremely desirable for proof of nanoscale materials to cite TEM data, in this case probably in its cryo-variant. Please take this remark into account in your further work with the manuscript.
Reply
- The term micelles is used to describe spherical structures that form when certain molecules, typically amphiphilic or surfactant molecules, are dispersed in a liquid, such as water. A nanoemulsion is a type of emulsion, which is a mixture of two immiscible liquids (usually oil and water) of small droplet size stabilized by an emulsifying agent or surfactant. In our work, the formulation contains stearic acid, cholesterol, and an oil (perrilyl alcohol or sefsol) which is different.
- We have modified the name of the prepared nanocarriers into lipid-based nanocarriers.
- TEM image for the the PA-loaded lipid-based nanocarriers’ formulation has been added.
- There is a rather large number of works devoted to the synthesis of nanostructured lipid carriers (NLCs) containing perillyl alcohol, but the authors do not raise this when discussing the current state of the problem. The authors should revise the introduction and present data on the synthesis of perillyl alcohol-based micelles and testing of the resulting materials as potential antitumor agents.
Reply
The introduction section in the revised manuscript has been modified to include the required information.
- The authors should unify terminology, in particular the spelling of their research objects (nanoparticles, nanodelivery agents, nanoemulsions, etc.).
Reply
We have unified terminology and used the term “nanocarriers” in the revised manuscript.
- All abbreviations (in particular, NLC) should be decoded at their first use in the text (with a dash, please).
Reply
The name of the prepared nanoformulations has been changed into lipid-based nanocarriers and unified in the revised manuscript. Also, the formulations have been assigned the following codes “NLC1, NLC2 and NLC3”. Abbreviations have been fully described at their first presence.
These changes have been implemented in the revised manuscript.
- The size of the order of 270-1120 nm does not allow to attribute the materials obtained by the authors to nanostructured materials.
Reply
Both formulation NLC1 and NLC2, which contain a solid lipid part and an oily part, showed a size of 330.47±22.11 and 248.67±12.42, respectively and are considered as nanostructured, while the formulation coded NLC3 that demonstrated a size of 1124.21±12.77 contains no oil. So, we changed the formulations’ name into lipid-based nanocarriers and kept the formulation codes.
- Micelle size was measured by DLS method at 25 C°. Whereas the cytotoxicity was measured at 37 °C. Taking into account the expressed potential of application of the obtained materials in biomedical fields it is necessary to be oriented on conditions of testing of their characteristics similar to biomimetic ones, in particular it is necessary to supplement the data on particle size, polydispersity index and zeta potential with measurements obtained at 37 °C.
Reply
- Measuring the size of nanoparticles, using Dynamic Light Scattering (DLS), at 25°C is often chosen because it's a temperature that allows for stable and consistent measurements. At this temperature, the nanocarriers are less likely to aggregate or undergo significant changes in size due to thermal motion, which could distort the measurements. Lab environments are often controlled at room temperature (around 25°C), making it a convenient choice for routine measurements. This helps ensure that measurements are repeatable and can be compared across different laboratories and experiments. Many studies and publications report DLS measurements at 25°C. Using this standard temperature allows for easier comparison and benchmarking of your nanocarriers' size data with existing literature and established data.
- Varenne et al (2015) published a paper related to Standardization and validation of a protocol of size measurements by dynamic light scattering for monodispersed stable nanomaterial characterization at 20 and 25 °C.
Fanny Varenne, Jérémie Botton, Claire Merlet, Moritz Beck-Broichsitter, François-Xavier Legrand, Christine Vauthier. Standardization and validation of a protocol of size measurements by dynamic light scattering for monodispersed stable nanomaterial characterization. Colloids and Surfaces A: Physicochemical and Engineering Aspects. Volume 486, 5 December 2015, Pages 124-138
- In biomedical applications, especially in studies related to cytotoxicity, it is crucial to mimic the conditions inside the human body as closely as possible. The human body maintains a temperature of around 37°C, and cells, including those used in cytotoxicity assays, function optimally at this temperature. Measuring cytotoxicity at 37°C provides results that are more biologically relevant and predictive of how the material will behave in the body.
- The authors should provide an explanation of the choice of Triton X as a positive control for the determination of the half maximal inhibitory concentration "IC50"
Reply
Triton X-100 was used as a positive control to disrupt the cell membrane through solubilization of membrane lipids and proteins, leading to cell lysis.
This explanation, supported by a reference, was added to the modified version of the manuscript.
- The authors should provide data on the stability of the resulting emulsions.
Reply
All the prepared nanocarriers showed a negatively charged particles surface ( -36.91±1.31 to -15.20±0.96 mV) due to the presence of stearic acid and cholesterol in all the prepared formulations . Although the prepared nanocarriers have a zeta potential values that are lower than the recommended value (should be >± 30), these values are sufficient to stabilize the prepared nanocarriers by both electrostatic and steric stabilization. Thermal and long-term storage stability testing are recommended to assess the behavior of these nanocarriers under different conditions.
The above explanation has been added to the revised manuscript
Comments on the Quality of English Language: A number of terms require checking of spelling.
Reply
We have improved the quality of English Language and corrected spelling mistakes.
Round 2
Reviewer 1 Report
The authors have sufficiently addressed my comments. However I still have some concerns about therapeutic window given transport across BBB and selective uptake concern. Do consider carefully for future studies.
Author Response
Reviewer 1
The authors have sufficiently addressed my comments. However I still have some concerns about therapeutic window given transport across BBB and selective uptake concern. Do consider carefully for future studies.
Reply
We acknowledge the importance of assessing the therapeutic window, especially when dealing with compounds for the treatment of brain malignancies. In our study, we aimed to evaluate the enhanced delivery of Perillyl Alcohol to glioma cells using lipid-based nanocarriers. While we have observed increased antiproliferative activity in glioma cells, we understand that further investigations are needed. In future studies, we plan to perform detailed pharmacokinetic and pharmacodynamic assessments, including the evaluation of BBB penetration, to better understand the compound's distribution and effects within the brain.
This explanation has been added to the revised manuscript.
Reviewer 3 Report
In the edited paper, the authors tried to take into account the comments submitted earlier, namely: they edited the introduction, added a TEM snapshot of the objects they obtained, and unified the spelling of their objects.
1) Figure 2 caption. The authors should correct the caption to Figure 2. Perhaps they meant that the presented figure is a TEM micrograph.
Author Response
Reviewer 3
1) Figure 2 caption. The authors should correct the caption to Figure 2. Perhaps they meant that the presented figure is a TEM micrograph.
Reply
Figure 2 caption has been corrected.